# Stakeholder acceptance and attitudes toward dengue prevention: A study on fogging and autocidal traps techniques in Malaysia

Noor Sharizad Rusly[1]*, Nur Asmadayana Hasim[1], Ahmad Firdhaus Arham[1], Latifah Amin[2]

**1** Pusat Pengajian Citra Universiti, Universiti Kebangsaan Malaysia, Bangi, Selangor, Malaysia, **2** Institut Islam Hadhari, Universiti Kebangsaan Malaysia, Bangi, Selangor, Malaysia

* n.sharizadrusly@ukm.edu.my

## Abstract

The ineffectiveness of current prevention efforts is underscored by the increasing number of dengue cases, emphasizing the necessity of innovative approaches. This investigation investigates the extent of stakeholder acceptability and attitudes toward the alternative autocidal trap method and the current fogging technique. The objective is to identify differences in community viewpoints regarding these control techniques among various stakeholder groups, including scientists and the public. Information was collected from two separate groups of individuals: scientists and the general people. A comprehensive survey assessed the level of tolerance and attitudes towards fogging and autocidal traps. Statistical analysis was performed using the SPSS software, specifically utilizing the one-way MANOVA test, to identify any significant variations in attitudes among various stakeholder groups. The results reveal significant discrepancies in the perceived danger level between the two groups since the public views fogging as a riskier activity than scientists. However, the groups had no significant difference in the reported benefits. Scientists exhibited a greater degree of endorsement for both fogging and autocidal traps when compared to the public. While the general population showed more openness towards fogging, scientists have shown a more balanced approval of both tactics. These insights are essential for policymakers and health practitioners as they inform the creation of more efficient, community-oriented initiatives to address the increasing prevalence of dengue cases.

## Introduction

Dengue fever, a viral infection spread by mosquitoes, is a significant global public health issue, especially in tropical and subtropical regions. It is among the most widespread vector-borne diseases and is rapidly proliferating. The disease is

**Data availability statement:** All relevant data are within the manuscript and its Supporting Information files.

**Funding:** The Ministry of Higher Education supported this research under the FRGS/1/2023/SSI03/UKM/02/1 and ERGS/1/2013/SSI12/UKM/02/1 grants. The funders had no role in study design, data collection and analysis, publication decision, or manuscript preparation.

**Competing interests:** The authors have declared that no competing interests exist.

predominantly transmitted by female Aedes aegypti mosquitoes (the principal vector) and Aedes albopictus mosquitoes (the secondary vector), with severe dengue resulting in a mortality rate of up to 20%. The number of dengue cases documented has risen tenfold over the last 20 years, from 505,430 cases in 2000 to over 5.2 million cases in 2019 [1]. Dengue fever is widespread in over 125 countries, impacting almost 3.6 billion individuals globally.

Despite dengue's severity and prevalence, there is currently no readily accessible vaccine or specialized antiviral cure for the illness [2]. Thus, vector management continues to be the principal technique for reducing dengue transmission [3]. Conventional preventive techniques, such as chemical fogging, are extensively employed to eradicate infected mosquito populations during epidemics. However, these tactics have proven poor long-term efficiency in preventing dengue transmission, as Aedes mosquitos have evolved pesticide resistance, and fogging alone is insufficient to terminate epidemics [4]. For dengue to keep spreading, only two to three mature female mosquitoes must hatch daily for every 100 people [5]; as little as two to three adults are needed.

Given the increasing prevalence of dengue cases, it is imperative to implement creative and sustainable vector control techniques. A promising method is the implementation of autocidal gravid ovitraps (AGOs), which have shown sustained efficacy in dengue-endemic regions [6,7]. AGOs are enhanced iterations of standard ovitraps intended to capture and eliminate mosquito larvae and gravid female mosquitoes, hence diminishing vector populations early [8]. Research done in Puerto Rico and Australia has shown that AGOs can effectively cut down on mosquito numbers by a large amount [6,7]. Nevertheless, there is still little widespread use of autocidal traps in South America, especially in Colombia, one of the most impacted nations in the area [9].

The efficiency of any dengue control plan is determined by scientific efficacy as well as public perception, acceptance, and stakeholder engagement. Studies demonstrate that community involvement markedly improves adherence and fosters the enduring viability of vector control initiatives [3]. The implementation of novel control methods is affected by various factors, such as perceived risks, perceived benefits, faith in public health authorities, and sociocultural influences [10]. Also, urban planning is very important for preventing dengue because Aedes aegypti mosquitoes love to breed in places with bad sewage systems, poor waste management, and many individuals [5].

Comprehending stakeholder perceptions of diverse dengue management strategies is crucial for formulating effective public health communication and engagement activities. Stakeholders, such as public health officials, community leaders, and local citizens, are crucial to the execution and efficacy of dengue preventive initiatives. Their apprehensions, awareness levels, and acceptance substantially influence the efficacy and durability of dengue prevention programs [2].

This study seeks to examine and contrast stakeholder acceptance and perspectives about fogging and autocidal traps as methods of dengue prevention. This research will use SPSS and one-way MANOVA analysis to investigate variances

in community views among different stakeholder groups. The results will offer significant insights into each strategy's perceived risks and advantages, subsequently informing the creation of more successful, community-oriented dengue preventive initiatives.

## Materials and methods

### Study design

The relationship between the variables listed in the framework of concepts was investigated in this study using a method based on quantitative research. This inquiry requires stakeholder viewpoints to create an inventory of attitudes and acceptance of dengue prevention and control measures using a survey. The sample size is determined according to [11] recommendation. To get a power value of 0.80 and identify a medium-sized impact ($f = 0.25$) with a significance level of $p = 0.05$ and a margin of error of $u = 10$, a sample size of 25 respondents is required for the target population. As shown in Table 1, the sample size of 415 ($n = 415$) is optimal for the study, as it is within the range of 384–500 samples. These samples are of people aged 18 and up and have been stratified according to the target category. Although our study employed a quantitative methodology to ensure objective and generalizable findings, we advise that future research use qualitative techniques such as focus groups or interviews to understand better stakeholder perceptions and the motivations, concerns, and contextual factors influencing public acceptance of dengue prevention measures.

### Instrument development and measures

The questionnaire utilized in this study was designed by an extensive examination of existing literature and thorough deliberation among the research team, which included field experts. To mitigate the possibility of bias that comes from data they reported, participants were provided with assurances of the confidentiality and privacy of their responses. The questionnaire was specifically created to mitigate survey weariness and was scrutinized by survey research professionals to ensure its face and content validity. The questionnaire was organized into four distinct sections. A field sample consisting of 415 responses was collected between 1 July 2017–31 December 2017 within the Klang Valley area. The questionnaires were administered both in English and Malay versions separately. All the respondents were informed about the survey's objectives and were required to give informed consent before participating in the survey. All information obtained from the pilot sample was excluded from subsequent analysis.

## Declaration

### Ethical approval and consent to participate

According to the Guidelines for the Ethical Review of Clinical Research or Research involving human subjects by the Medical Review and Ethics Committee (MREC), Ministry of Health Malaysia [28], ethical permission was not necessary for this study. The research would be exempt from MREC approval because it is a study that utilizes surveys to investigate public behaviour without gathering any personally identifiable information. MREC may also exempt the need for individual informed consent if subjects are exposed to minimal risk and the study solely involves publicly accessible data. Nevertheless, before answering the questionnaire, all respondents were orally provided with informed consent. The respondents'

**Table 1. Respondents' demographic background.**

| Demographic background | Frequency | Percentage |
|---|---|---|
| Stakeholders | | |
| Scientist | 212 | 51.1 |
| Public | 203 | 48.9 |

participation was optional, and they could quit at any time. Upon consenting to complete this questionnaire, just the respondent's unique identification number and the completion date are recorded on the front of the survey paper, devoid of any personal details. The participants responded to this survey in person, directly engaging with the researchers and enumerators to guarantee that all questions were promptly addressed.

## Acceptance of dengue alternative techniques

The participants were surveyed about their willingness to adopt the alternative strategy of using autocidal traps to battle dengue. They were given a scale of 7 response levels, ranging from "very strongly disagree" to "very strongly agree". The variables examined as possible predictors of acceptability of dengue prevention techniques were stakeholders, including scientists and the public.

## Attitudes toward dengue alternative techniques

The survey on attitudes towards alternate methods of preventing dengue consists of 24 items (Table 2). This study adopted a 7-point Likert scale to evaluate the participants' attitudes and perspectives concerning fogging and autocidal traps as approaches for dengue prevention. The Likert scale offers a systematic method for gathering opinions, allowing respondents to express their degree of agreement or acceptance of a certain assertion. Frequently employed in social and health sciences research, this scale facilitates a more sophisticated assessment of attitudes than a basic binary response system [12,13].

Responses on the 7-point Likert scale are organized as follows: 1 (Strongly Disagree), 2 (Disagree), 3 (Somewhat Disagree), 4 (Neutral), 5 (Somewhat Agree), 6 (Agree), 7 (Strongly Agree). By using this method, researchers can examine subtle changes in attitudes instead of categorizing respondents into strict "agree/disagree" groups [14]. In order to evaluate the average scores from Likert scale replies, the study divides the findings into three acceptance levels: low,

Table 2. Items of the instrument.

| Factor | Total items | Items | Sources |
|---|---|---|---|
| Perceived benefit | 8 items | 1. will enhance the quality of Malaysian society life<br>2. and be useful to the Malaysian society<br>3. useful in preventing the outbreak of dengue fever<br>4. effectives to eradicate dengue<br>5. beneficial to me and my family's health.<br>6. benefits to people outweigh their risks.<br>7. will be dealt with future research.<br>8. scientific evidence on the safety is adequate. | [23–26,28,29] |
| Perceived risk | 10 Items | 9. pose harm to the ecosystem and environment.<br>10. indirectly reducing earth's biodiversity.<br>11. harm other organisms in ways that we do not know.<br>12. the level of your anxiety about the unknown effects.<br>13. nny harmful effect will only manifest itself after long term duration?<br>14. harm the next generation.<br>15. may give rise to unknown consequences.<br>16. may cause a major catastrophe to Malaysian society.<br>17. the level of your anxiety about the potential risks<br>18. adverse effects are harmful. | [26–29] |
| Acceptance | 6 items | 19. willing to support if it can combat dengue.<br>20. willing to support if it is beneficial to my health and the health of people important to me.<br>21. willing to support the if there is no better alternative.<br>22. willing to support if they have a potential to combat dengue effectively.<br>23. willing to support if they have been proven effective to combat dengue in other areas.<br>24. willing to support if the government can ensure its effectiveness. | [23–29] |

moderate, and high. This classification aids in recognizing patterns in stakeholder perspectives about the two vector control techniques. According to prior studies in public health behavior [15,16], the division adheres to these computed mean score ranges:

Low Acceptance (1.00–3.00): Responses within this range signify strong disagreement, disagreement, or mild disagreement, indicating a low level of acceptance for the assessed approach or tactic. A mean score within this range indicates that respondents view the intervention as ineffective, superfluous, or adverse [17]

Moderate Acceptance (3.01–5.00): This range encompasses neutral or partially agreeable responses, indicating ambivalence or conditional endorsement of the strategy. Individuals within this category may recognize the method's advantages, but they remain sceptical regarding its overall efficacy or practical implementation [18]

High Acceptance (5.01–7.00): A mean score within this range signifies agreement to strong agreement, reflecting substantial acceptance and confidence in the procedure. Research indicates that elevated acceptance scores are associated with favorable behavioral intentions and readiness to embrace public health interventions [19]

Implementing a 7-point scale and the classification into three acceptability levels enhances measurement precision relative to a 5-point scale, facilitating increased variation and diminishing central tendency bias [20]. This method is especially useful in public health research because precise measurement of behavioral intentions and attitudes toward interventions is necessary to formulate effective policy recommendations [2,21]. This classification enables the study to delineate stakeholder views regarding fogging and autocidal traps, hence enhancing comprehension of the elements affecting public acceptance of vector management technologies. The outcomes obtained from this methodology can enhance public health communication strategies and intervention planning, ensuring that vector control initiatives correspond with stakeholder viewpoints and behavioral motivators.

## Statistical analysis

Descriptive analysis using SPSS was initially employed to ascertain the mean score for each statement according to respondents' selections. After obtaining the mean results, a three-tier categorization approach was utilized to evaluate the level of acceptance of dengue prevention and control strategies. Responses were categorized into low, moderate, or high acceptance levels, facilitating a systematic assessment of stakeholder viewpoints [22]. This classification offers a systematic, objective, and dependable framework for evaluating attitudes toward dengue prevention strategies. After the descriptive analysis, a one-way multiple analysis of variance (MANOVA) was performed to investigate significant differences in acceptability and attitudes regarding dengue prevention techniques among various stakeholder groups in the Klang Valley. Upon a significant discrepancy reported by the MANOVA, post-hoc tests were used to ascertain group differences. A significance level 0.05 was employed for all MANOVA and post-hoc analyses, assuring statistical validity. This method thoroughly evaluates perceptual differences regarding adopting dengue prevention and control techniques among essential stakeholder groups, such as scientists and the public.

## Demographics

After the recollection process, 415 questionnaires were obtained. According to the data presented in Table 1, 415 replies were gathered, with 203 (48.9 percent) from the public and 212 (51.1 percent) from the group of scientists.

## Perceived benefits

The analysis reveals that both main stakeholder groups, scientists and the public, exhibit considerable interest in fogging techniques and autocidal traps, as illustrated in Table 3. Among the two dengue prevention approaches, autocidal traps garnered the greatest perceived benefit score, with scientists averaging 5.41 and the public 5.37. The general people

**Table 3. Mean score and standard deviation analysis of perceived benefit, perceived risk, and acceptance for fogging technique and autocidal trap.**

| Prevention technique | Stakeholder | Mean score±Std deviation | Interpretation |
|---|---|---|---|
| Fogging | Perceived benefit | | |
| | Scientist | 5.20±1.27 | High |
| | Public | 5.29±1.10 | High |
| | Perceived risk | | |
| | Scientist | 2.30±0.76 | Low |
| | Public | 2.17±0.67 | Low |
| | Acceptance | | |
| | Scientist | 5.80±1.11 | High |
| | Public | 5.67±1.13 | High |
| Autocidal trap | Perceived benefit | | |
| | Scientist | 5.41±1.13 | High |
| | Public | 5.37±0.93 | High |
| | Perceived risk | | |
| | Scientist | 3.72±1.71 | Moderate |
| | Public | 3.64±1.36 | Moderate |
| | Acceptance | | |
| | Scientist | 5.85±1.08 | High |
| | Public | 5.68±0.99 | High |

regarded fogging as marginally less advantageous, with an average score of 5.29, whereas scientists assessed it at 5.20. A one-way multiple analysis of variance (MANOVA) was performed to evaluate the statistical differences in perception between the two stakeholder groups. The findings (Table 4) indicated a statistically significant disparity in the perceived advantages of fogging preventive measures (F = 4.407, $p < 0.05$; Pillai's Trace = 0.031) and autocidal traps (F = 3.420, $p < 0.05$; Pillai's Trace = 0.02). These results show that, at the multivariate level, stakeholder perceptions of the two dengue prevention strategies vary substantially. Nonetheless, when examined individually through univariate tests, the findings indicate no statistically significant difference in the average perception of benefits between fogging and autocidal traps. This indicates that although the stakeholder groups exhibit considerable variation in perception at the multivariate level, their general sentiments regarding the efficacy and advantages of both strategies remain largely consistent. This organized analysis of the findings offers an in-depth comprehension of stakeholder perceptions, facilitating a more distinct comparison of attitudes between scientists and the public regarding dengue prevention efforts.

## Perceived risk

The analysis indicates that both scientists and the public regard fogging as posing a low, negligible risk and societal impact, as evidenced by the low mean scores of 2.30 (scientists) and 2.17 (public) (Table 3). The data indicate that stakeholders predominantly perceive fogging as a low-risk strategy for dengue prevention. The analysis indicates that

**Table 4. One-way MANOVA analysis on attitudes towards fogging and the autocidal trap technique across stakeholders.**

| Prevention technique | Dependent variables | Pillai's trace values | F value | Hypothesis.df | Error.df | Sig. |
|---|---|---|---|---|---|---|
| Fogging | Stakeholder | 0.031 | 4.407 | 3 | 411 | 0.005* |
| Autocidal trap | | 0.02 | 3.420 | 3 | 411 | 0.017* |

*$p < 0.05$

both scientists and the public regard fogging as posing negligible risk and societal impact, as evidenced by the low mean scores of 2.30 (scientists) and 2.17 (public) (Table 3). The data indicate that stakeholders predominantly perceive fogging as a low-risk strategy for dengue prevention, exhibiting less apprehension over its possible adverse effects. The perceived risk of implementing autocidal traps was moderate for both stakeholder groups. Researchers indicated a mean risk perception score of 3.72, although the public assessed it somewhat lower at 3.64. This moderate score indicates a degree of apprehension or ambiguity concerning the hazards associated with this technique, especially among scientists. A statistically significant difference in the average risk perception score for autocidal traps was found using a unidirectional univariate test to investigate these differences (F(1,413) = 7.66, p = 0.006) (Table 5). The findings demonstrate that scientists possess a markedly elevated risk perception of autocidal traps compared to the general populace. Nevertheless, a comparison of risk perception levels for fogging revealed no significant difference between the two stakeholder groups (Table 4). These results demonstrate a clear disparity in stakeholder perceptions, with fogging being regarded as a low-risk intervention and autocidal traps, especially among scientists, being viewed with mild caution. This understanding is crucial for formulating public health messaging and risk communication strategies, ensuring that both scientific and public concerns are sufficiently addressed in dengue prevention initiatives.

## Acceptance

The mean scores from scientists and the public were high, with 5.80 and 5.67 for fogging and 5.85 and 5.68 for autocidal traps, respectively. This demonstrates their endorsement and assistance to all stakeholders in implementing the strategies for preventing and managing dengue. A univariate one-way test indicated that there was no statistically significant disparity in the average score across all dengue prevention and control methods (Table 5).

Table 5. One-way MANOVA (univariate) test on attitudes towards fogging and autocidal trap techniques.

| Prevention techniques | Independent variable | Effect | sum of sqr | df | Square mean | F value | Sig. |
|---|---|---|---|---|---|---|---|
| Fogging | Perceived benefit | Stakeholders | 0.116 | 1 | 0.116 | 0.074 | 0.785 |
| | | Error | 644.242 | 413 | 1.560 | | |
| | | Total | 12125.36 | 415 | | | |
| | Perceived risk | Stakeholders | 1.748 | 1 | 1.748 | 3.395 | 0.066 |
| | | Error | 212.622 | 413 | 0.515 | | |
| | | Total | 2294.87 | 415 | | | |
| | Acceptance | Stakeholders | 1.808 | 1 | 1.808 | 1.445 | 0.230 |
| | | Error | 516.60 | 413 | 1.251 | | |
| | | Total | 14167.6 | 415 | | | |
| Autocidal trap | Perceived benefit | Stakeholders | 0.67 | 1 | 0.67 | 0.53 | 0.466 |
| | | Error | 518.60 | 413 | 1.26 | | |
| | | Total | 12685.56 | 415 | | | |
| | Perceived risk | Stakeholders | 18.37 | 1 | 18.37 | 7.66 | 0.006* |
| | | Error | 990.26 | 413 | 2.40 | | |
| | | Total | 7182.44 | 415 | | | |
| | Acceptance | Stakeholders | 3.14 | 1 | 3.14 | 2.90 | 0.089 |
| | | Error | 446.90 | 413 | 1.08 | | |
| | | Total | 14260.25 | 415 | | | |

*p < 0.02

## Discussion

Dengue fever is still a major world public health problem that needs comprehensive and flexible control strategies to stop the spread. Conventional vector control techniques, such as fogging, have been extensively utilized to eradicate adult Aedes mosquitoes. Nonetheless, their effectiveness is frequently temporary and significantly influenced by environmental and operational variables [8,10]. Although fogging is acknowledged for its rapid effects, research demonstrates that it does not yield lasting decreases in mosquito populations, highlighting the want for alternate and supplementary control strategies [2].

Dengue transmission's intricacy requires comprehensive management strategies, as outbreaks are affected by urbanization, socioeconomic variables, and environmental circumstances [5]. The Klang Valley region in Malaysia, responsible for 57% of the nation's dengue infections, encounters significant issues stemming from fast urbanization, ineffective waste management, and insufficient drainage systems, hence fostering optimal breeding conditions for Aedes mosquitoes [30]. Investigations into low-income neighborhoods in Klang Valley reveal that inadequate housing conditions, inappropriate water storage, and inefficient drainage systems facilitate mosquito reproduction, underscoring the need for comprehensive dengue control techniques [30].

The public's perspective is crucial for the adoption and efficacy of dengue prevention strategies. Stakeholder acceptance is significantly affected by perceived benefits and risks, which influence public participation in control activities [31,32]. Research in Klang Valley reveals that participants recognize the benefits of both fogging and autocidal traps; nevertheless, scientific evaluations indicate that autocidal traps (mean score: 5.41) are seen as more effective than fogging (5.26) [33]. This underscores the imperative for technology-driven treatments, evidenced by heightened public acceptance when a method's advantages are thoroughly established.

Nonetheless, perceived risks—such as financial liabilities, health issues, and uncertainties—hinder the implementation of innovative techniques [31]. Participants in Klang Valley typically view fogging and autocidal traps as advantageous; however, scientists contend that autocidal traps provide superior long-term effectiveness, hence affecting public approval. Research demonstrates that perceived benefits significantly affect stakeholder endorsement of biotechnology applications, but perceived risks—such as possible economic detriments or health issues—can diminish public confidence in innovative solutions [34,35].

When evaluating dengue prevention strategies, perceptions of risk and benefit are linked. A heightened sense of danger frequently corresponds with diminished trust, underscoring the necessity for public education and awareness initiatives [31]. A study conducted in Seremban, Malaysia, revealed that risk perception modulates the association between public attitudes and dengue preventative practices, underscoring the necessity of focused interventions [36,37]. Research further substantiates that behavioral change initiates with risk awareness, as individuals are more inclined to alter health practices when perceiving a tangible threat [38]. In Singapore, where the population was more aware of the risks, they were more likely to take precautions [39].

Knowledge and experience profoundly influence attitudes and behaviors regarding fogging and autocidal traps, as both approaches necessitate public comprehension of their ecological and vector control roles. Research demonstrates that heightened awareness of dengue prevention is associated with enhanced attitudes and behaviors [40]. Educational initiatives targeting mosquito breeding patterns and daytime biting behaviors have demonstrated increased efficacy of fogging and autocidal traps, especially within university environments [40]. Likewise, [41] discovered that public choices regarding the utilization of fogging and autocidal traps are shaped by their comprehension of Aedes mosquito behavior and socioenvironmental variables.

The ecological importance of maintaining a balanced environment as a vector control method corresponds with research highlighting the effects of urbanization, waste management, and water storage on mosquito population growth [41]. According to the findings of a cross-sectional study conducted by [42], there is a substantial association between knowledge concerning dengue and positive attitudes toward preventive measures. Increasing awareness can

lead to increased community engagement. In addition to technical effectiveness, public knowledge of the environmental and health advantages of fogging and autocidal traps is essential for successful deployment and long-term sustainability [40,41].

Integrating education and practical demonstrations is crucial for enhancing public acceptance of modern vector control technology. The future should include community-oriented educational initiatives to rectify misconceptions and encourage the implementation of innovative dengue prevention methods. The efficacy of vector control techniques depends on stakeholder involvement, public awareness, and the incorporation of evidence-based interventions.

Notwithstanding progress in vector control, mosquito suppression continues to be the principal tactic due to the lack of a broadly accessible dengue vaccine or antiviral therapy (WHO, 2024). Nonetheless, efficient vector control initiatives might diminish herd immunity, requiring an additional decrease in mosquito populations to sustain balance and avert rebound [43]. The perceived benefits are the most important factor in determining how the general population feels about dengue prevention. Participants in Klang Valley exhibit more acceptance of tactics that present evident benefits, corroborating findings from research on genetically modified mosquitoes and biodiesel applications, where perceived advantages were a primary determinant of public endorsement [44–46].

Prevalent perceptions of dengue prevention differ according to regional and cultural influences. Significant support for the dengue vaccine was noted in Bandung, Indonesia, as the perceived communal advantages surpassed the worries [47]. Similarly, in Kaohsiung City, Taiwan, the combination of aerosol pesticide sprays and domestic ovitraps contributed to a 60% decrease in dengue infections, highlighting the importance of aligning prevention efforts with public attitude [48].

However, this study has certain analytical and methodological benefits. Comparisons between the public and scientific groups were made methodically using validated categories and composite mean scores to assess stakeholder views of fogging and autocidal traps. The application of MANOVA enabled the simultaneous assessment of many dependent variables, improving the effectiveness and robustness of group comparisons. The sample size (n = 415) provided considerable statistical power to detect significant differences, while segmentation by stakeholder group ensured thorough representation of public and scientific perspectives.

Nonetheless, specific limitations must be acknowledged. The study employed non-randomized groupings, limiting the ability to reduce selection bias in stakeholder comparisons. Although the statistical assumptions for MANOVA have been satisfied through testing, the interpretation of Likert-scale data may still provide issues, as it is regarded as an interval for parametric analysis. Moreover, due to its cross-sectional design, the findings reflect perspectives at a specific time and do not account for changes in acceptance or awareness over time. Despite these limitations, the findings provide a substantial overview of stakeholder perspectives and may inform the development of future public health communication and engagement methods.

## Conclusion

This study emphasizes the essential influence of public perception on the implementation and efficacy of dengue prevention efforts. Although fumigation is still broadly endorsed, apprehensions surrounding pesticide resistance, health hazards, and environmental contamination underscore the necessity for alternative and supplementary approaches, including autocidal traps and biological control methods. Research reveals that stakeholder perceptions of advantages and hazards substantially affect acceptance rates, with increased public endorsement when a method's efficacy is well established. The efficacy of dengue prevention efforts depends on public acceptance and adherence. Both conventional fogging and innovative autocidal traps necessitate substantial acceptance and favorable attitudes for effective use. When individuals acknowledge the advantages of preventative actions, such as eradicating mosquito breeding areas and implementing protective methods, they are more inclined to adopt and maintain these activities. This study investigated stakeholder perceptions of fogging and autocidal traps, demonstrating that perceived advantages and hazards substantially influence

adoption and effectiveness. The results underscore the necessity of a comprehensive dengue prevention strategy incorporating several control measures to improve overall efficacy and reduce disease burden. A comprehensive, evidence-based strategy is crucial for dengue's sustainable and enduring prevention. Dengue prevention efforts can be greatly improved by combining public health measures, personal safety measures, and community-led actions. The amalgamation of scientific progress, public awareness campaigns, and environmental stewardship is essential for enhancing the sustainability and effectiveness of vector control methods. The efficacy of these treatments relies on cultivating favorable public impressions while mitigating concerns regarding safety, effectiveness, and accessibility. The success of dengue control initiatives depends on raising public awareness, fostering confidence in emerging technology, and promoting community involvement. This study highlights the importance of evidence-based public health strategies prioritizing community engagement and integrated approaches. Enhancing public comprehension and encouraging active engagement will foster a more resilient and proactive society, so effectively diminishing dengue transmission and protecting public health. Subsequent studies ought to investigate the socioeconomic and behavioral determinants affecting the uptake of innovative vector control methods, establishing a basis for evidence-based public health policies. By combining new scientific discoveries with active community participation, dengue prevention efforts can be more successful and last longer, reducing the impact of dengue fever worldwide.

## Supporting information

**S1 File. Questionnaire response dataset on stakeholder acceptance and attitudes toward fogging and autocidal trap techniques for dengue prevention in Malaysia.**
(XLSX)

## Author contributions

**Conceptualization:** Noor Sharizad Rusly.

**Data curation:** Noor Sharizad Rusly.

**Formal analysis:** Noor Sharizad Rusly, Ahmad Firdhaus Arham.

**Funding acquisition:** Ahmad Firdhaus Arham.

**Methodology:** Noor Sharizad Rusly, Ahmad Firdhaus Arham.

**Software:** Ahmad Firdhaus Arham.

**Supervision:** Noor Sharizad Rusly, Latifah Amin.

**Visualization:** Noor Sharizad Rusly.

**Writing – original draft:** Noor Sharizad Rusly.

**Writing – review & editing:** Nur Asmadayana Hasim, Ahmad Firdhaus Arham.

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
