## [Decision Letter · Decision Letter 0]

PONE-D-24-26520Stakeholder Acceptance and Attitudes Toward Dengue Prevention: A Study on Fogging and Autocidal Traps Techniques in Malaysia.PLOS ONE

Dear Dr. Rusly,

Thank you for submitting your manuscript to PLOS ONE. After careful consideration, we feel that it has merit but does not fully meet PLOS ONE’s publication criteria as it currently stands. Therefore, we invite you to submit a revised version of the manuscript that addresses the points raised during the review process.

We look forward to receiving your revised manuscript.

Kind regards,

Rajib Chowdhury, M.Sc.; MPH

Academic Editor

PLOS ONE

Journal Requirements:

[FRGS/1/2023/SSI03/UKM/02/1 THIS GRANT IS FUNDED BY MINISTRY OF HIGHER EDUCATION].

Please confirm at this time whether or not your submission contains all raw data required to replicate the results of your study. Authors must share the “minimal data set” for their submission. PLOS defines the minimal data set to consist of the data required to replicate all study findings reported in the article, as well as related metadata and methods (https://journals.plos.org/plosone/s/data-availability#loc-minimal-data-set-definition ).

If your submission does not contain these data, please either upload them as Supporting Information files or deposit them to a stable, public repository and provide us with the relevant URLs, DOIs, or accession numbers. For a list of recommended repositories, please see https://journals.plos.org/plosone/s/recommended-repositories .

5. Please update your submission to use the PLOS LaTeX template. The template and more information on our requirements for LaTeX submissions can be found at http://journals.plos.org/plosone/s/latex .

Reviewers' comments:

Reviewer's Responses to Questions

**Comments to the Author**

1. Is the manuscript technically sound, and do the data support the conclusions?

Reviewer #1: Yes

Reviewer #2: Yes

Reviewer #3: Yes

2. Has the statistical analysis been performed appropriately and rigorously? 

Reviewer #1: Yes

Reviewer #2: Yes

Reviewer #3: Yes

3. Have the authors made all data underlying the findings in their manuscript fully available?

Reviewer #1: Yes

Reviewer #2: Yes

Reviewer #3: No

4. Is the manuscript presented in an intelligible fashion and written in standard English?

Reviewer #1: Yes

Reviewer #2: Yes

Reviewer #3: Yes

5. Review Comments to the Author

Reviewer #1: In this manuscript I evaluate the proposal for an alternative selective chemical control with proven effectiveness in previous studies and the contribution that complements its feasible possibility in its useful application, with acceptance and possible participation of involved actors, including the population exposed to the problem. The validation of the measurement questionnaire, data collection, analysis and results were prepared with sufficient methodological rigor.

Reviewer #2: 1.- Relevance in the Introduction:

It is important to highlight the importance of health promotion in relation to the increase in patients with dengue in localities where there was previously no presence of autochthonous cases, that is, the increase in endemic localities. In seeking economically and operationally viable solutions, public opinion from different sectors plays a transcendental role in the effectiveness of dengue prevention and control work, that is, intersectoral opinion and participation.

Evaluating the acceptance of different control techniques, when perhaps a change in the operational activities of a particular area is intended, is crucial, as is evaluating the parameters in each of the localities, since biotic and abiotic conditions, relief, population density, ambient temperature, annual rainfall, among others, largely determine what type of actions should be implemented in each locality, taking into account its own challenges in the operational work that is desired to be implemented.

Lack of knowledge about the disease itself, as well as about the activities implemented in the population, could cause problems or delays in carrying out the activity, due to the fact that a certain number of variables to be implemented are not considered, as well as making the population aware of the new techniques to be implemented. In addition to this, the health culture in each of the regions to be established greatly favors or hinders the impact of one or several strategies.

As mentioned, the rapid expansion of the disease, together with the adaptability of the vectors to new environments, that is, environments where the disease was not common, can be associated with the effectiveness of the different activities carried out for prevention and control of dengue. Other factors that largely determine the presence of potential breeding sites are the urban planning of the same areas.

It is important to include new innovations in control, as well as the evaluation of acceptance in the population in each of the different regions. Allowing the experimentation of different techniques based on monitoring by ovitraps is essential to improve the initial versions of this methodology, that is, the more new designs are allowed to be experimented with in the autocidal ovitraps, the more efficiently they will be able to achieve the objective of reducing the density of the vector.

Including the participation of the different institutions, both from the public and private educational sectors, the environmental sector, institutions related to the supply of drinking water, institutions related to the collection of urban solid waste, public and private health institutions, research institutions as well as local and federal governments of each of the nations, added together in a single objective which is the reduction of dengue cases. Likewise, the creation of public policies in each of the local governments could help to pressure for the creation of new strategies that help to mitigate this condition.

2.- Adequate methodology

The quantitative measurement in the perspectives of attitudes and acceptance of measures for the prevention of dengue is adequate, it could be mentioned that it is also a qualitative research, since it is focused on determining the opinion of the different target populations. It is very important to highlight this agreement of the acceptance of a methodology in both groups, to ensure the population benefits from the new activity to be implemented.

3.- Discussion

All activities focused on preventing dengue are important for public health, it is important to mention the average score of the response of the most significant benefit received from nebulization, which gives rise to implementing new control and prevention measures, but it will be the sum of many strategies that determine the optimal impact of the reduction of this disease. This is focused on the fact that depending on the population at risk in a given area, as well as the environmental variables, the type of methodology to be implemented will be determined. One could mention the type of variables that exist in the Klang Valley in Malaysia, that is, the socioeconomic and environmental variables.

Likewise, the lack of knowledge of the disease in many areas can change the entire methodology of a prevention and control activity, as it is mentioned that 38.8% of those surveyed are not sure that the problem can be solved. It is very good that they mention the health benefits of their community and their quality of life.

Reviewer #3: 1. The primary aim of this study was to investigate and contrast the level of acceptance and attitudes of stakeholders towards fogging and autocidal traps as methods for preventing dengue fever.

Comment :

This study does not explain in more detail the factors that influence a person's attitude and decision-making such as age, education, experience, social and cultural environment which is actually a matter of principle to describe the existing research population. Maybe this should be added to the data.

2. In study design : These samples represent people aged 18 years and above, and they have been stratified based on the target group. Thestakeholders, consisting of scientists and the public, were divided into sub-groups based on the study site within the Klang Valley, Malaysia, specifically the districts and municipalities. The scientists were stratified by universities and departments, including ministries, institutes and health offices. The public was doing sampling in accordance with dengue hotspot regions mentioned on the i-dengue website provided by the Ministry of Health,Malaysia.

Comment : Can you more explain how many group you have been stratified ? There are no data for . The scientists and publics were stratified .

3. In Acceptance of dengue alternative techniques : They were provided with a scale of 7 response levels, ranging from "very strongly disagree" to "very strongly agree".

Comment : Please explain in detail the scale division of the 7 response levels of the limit in the selection of the answer option ?

4. The survey on attitudes towards alternate methods of preventing dengue consists of 24 items with Participants rate their agreement on a 7-point Likert scale, with 7 indicating strong agreement and 1 indicating strong disagreement.

Comment : Please explain in detail the scale division of the 7 point Likert scale and the limit in the selection of the answer option ?

5. Perceived risk and benefit are not explained in terms of grouping and analysis

6. In the discussion it is not explained in more detail that knowledge of fogging and autocidal traps is greatly influenced by knowledge and experience in taking attitudes. Knowledge of the importance of environmental balance as an ecological place where disease vectors live will determine a person's choice to make decisions in preventive measures against vector-borne diseases so that environmental control becomes an important thing to be raised as disease prevention in addition to treatment

7. The conclusion of the results should further explain the purpose of the study

6. PLOS authors have the option to publish the peer review history of their article (what does this mean? ). If published, this will include your full peer review and any attached files.

**Do you want your identity to be public for this peer review?** For information about this choice, including consent withdrawal, please see our Privacy Policy .

Reviewer #1: No

Reviewer #2: **Yes: ** Daniel Antonio Escobedo Torres

Reviewer #3: **Yes: ** Janno Berty Bradly Bernadus

---

## [Author Response · Author response to Decision Letter 1]

3 Mar 2025

We really value the reviewers' and editors' comments on our work.

We affirm that our manuscript is methodologically robust, featuring comprehensive statistical analysis, suitable methodologies, adequate controls, and sufficient sample sizes to substantiate our findings. We have verified that all statistical analyses were performed correctly and are reported with precision.

According to PLOS Data Policy, we provided summary statistics, means, medians, and variance measurements in supporting information files in a public repository. Should Reviewer #3 possess particular concerns, we invite further elucidation and are prepared to furnish additional information as required.

With regard to clarity and language, we have ensured that the manuscript is articulated in standard English with precision. We have meticulously reviewed and rectified typographical or grammatical inaccuracies to improve readability and precision.

Furthermore, we have addressed all feedback and comments provided by the three reviewers and made the necessary amendments accordingly. A comprehensive, point-by-point response addressing each reviewer comment, delineating the revisions undertaken, has been included in the rebuttal letter entitled "Response to Reviewers" within the specified section.

We are grateful for the reviewers' insightful contributions, which have significantly enhanced the manuscript. We remain receptive to any additional suggestions for improvement, should they be necessary. We appreciate your time and consideration.

---

## [Decision Letter · Decision Letter 1]

PONE-D-24-26520R1Stakeholder Acceptance and Attitudes Toward Dengue Prevention: A Study on Fogging and Autocidal Traps Techniques in Malaysia.PLOS ONE

Dear Dr. Rusly,

Thank you for submitting your manuscript to PLOS ONE. After careful consideration, we feel that it has merit but does not fully meet PLOS ONE’s publication criteria as it currently stands. Therefore, we invite you to submit a revised version of the manuscript that addresses the points raised during the review process.

We look forward to receiving your revised manuscript.

Kind regards,

Rajib Chowdhury, M.Sc.; MPH

Academic Editor

PLOS ONE

Reviewers' comments:

Reviewer's Responses to Questions

**Comments to the Author**

1. If the authors have adequately addressed your comments raised in a previous round of review and you feel that this manuscript is now acceptable for publication, you may indicate that here to bypass the “Comments to the Author” section, enter your conflict of interest statement in the “Confidential to Editor” section, and submit your "Accept" recommendation.

Reviewer #1: All comments have been addressed

Reviewer #2: All comments have been addressed

Reviewer #4: (No Response)

2. Is the manuscript technically sound, and do the data support the conclusions?

Reviewer #1: Yes

Reviewer #2: Yes

Reviewer #4: Partly

3. Has the statistical analysis been performed appropriately and rigorously? 

Reviewer #1: Yes

Reviewer #2: Yes

Reviewer #4: No

4. Have the authors made all data underlying the findings in their manuscript fully available?

Reviewer #1: Yes

Reviewer #2: Yes

Reviewer #4: No

5. Is the manuscript presented in an intelligible fashion and written in standard English?

Reviewer #1: Yes

Reviewer #2: Yes

Reviewer #4: Yes

6. Review Comments to the Author

Reviewer #1: The authors of the scientific work have responded with rigor and clarity to the questions and recommendations of the manuscript. I consider that this is a publication that meets the criteria of the journal. Its publication contributes with strong evidence to guide public health decisions for the sustainable prevention of diseases transmitted by Aedes spp. mosquitoes.

Reviewer #2: (No Response)

Reviewer #4: This is a good descriptive presentation. However, the inferential statistical approach is not necessarily robust as claimed and needs justification. Also, further detail should be considered as follows:

Please justify the MANOVA analysis since this is likert type data. Why not a non parametric type of analysis or other approach for this kind of data? Was the data checked for normality?

An expanded statistical analysis section is needed for all primary and secondary analysis endpoints.

Also the stakeholder groups are not randomized which calls into question the MANOVA approach. Was a propensity matching considered for the two groups since formal comparative statistical tests are being performed on a non randomized sample. The sample size looks large enough to tabulate the demographics, if any, were selected in the data collection.

The Discussion section needs to list the advantages and limitations encountered in this research.

7. PLOS authors have the option to publish the peer review history of their article (what does this mean? ). If published, this will include your full peer review and any attached files.

**Do you want your identity to be public for this peer review?** For information about this choice, including consent withdrawal, please see our Privacy Policy .

Reviewer #1: **Yes: ** Ángel Betanzos Reyes

Reviewer #2: No

Reviewer #4: No

---

## [Author Response · Author response to Decision Letter 2]

25 Jun 2025

Thank you for your valuable feedback and suggestions from a Reviewer #4. We have addressed all the points raised and revised the manuscript accordingly.

The detailed justifications and amendments have been included in the attached Response to Reviewers document for your kind consideration.

---

## [Decision Letter · Decision Letter 2]

Stakeholder Acceptance and Attitudes Toward Dengue Prevention: A Study on Fogging and Autocidal Traps Techniques in Malaysia.

PONE-D-24-26520R2

Dear Dr. Rusly,

We’re pleased to inform you that your manuscript has been judged scientifically suitable for publication and will be formally accepted for publication once it meets all outstanding technical requirements.

Kind regards,

Rajib Chowdhury, M.Sc.; MPH

Academic Editor

PLOS ONE

Additional Editor Comments (optional):

Reviewers' comments:

Reviewer's Responses to Questions

**Comments to the Author**

1. If the authors have adequately addressed your comments raised in a previous round of review and you feel that this manuscript is now acceptable for publication, you may indicate that here to bypass the “Comments to the Author” section, enter your conflict of interest statement in the “Confidential to Editor” section, and submit your "Accept" recommendation.

Reviewer #1: All comments have been addressed

Reviewer #4: All comments have been addressed

2. Is the manuscript technically sound, and do the data support the conclusions?

Reviewer #1: Yes

Reviewer #4: (No Response)

3. Has the statistical analysis been performed appropriately and rigorously? 

Reviewer #1: Yes

Reviewer #4: (No Response)

4. Have the authors made all data underlying the findings in their manuscript fully available?

Reviewer #1: Yes

Reviewer #4: (No Response)

5. Is the manuscript presented in an intelligible fashion and written in standard English?

Reviewer #1: Yes

Reviewer #4: (No Response)

6. Review Comments to the Author

Reviewer #1: El manuscrito es adecuado a los criterios de la editorial, no tengo recomendaciones adicionales. Médico epidemiólogo con experiencia en dengue y otras arbovirosis en México.

Reviewer #4: (No Response)

7. PLOS authors have the option to publish the peer review history of their article (what does this mean? ). If published, this will include your full peer review and any attached files.

**Do you want your identity to be public for this peer review?** For information about this choice, including consent withdrawal, please see our Privacy Policy .

Reviewer #1: **Yes: ** Angel Francisco Betanzos Reyes

Reviewer #4: No

---

## [Editor Report · Acceptance letter]

PONE-D-24-26520R2

PLOS ONE

Dear Dr. Rusly,

I'm pleased to inform you that your manuscript has been deemed suitable for publication in PLOS ONE. Congratulations! Your manuscript is now being handed over to our production team.

Kind regards,

on behalf of

Dr. Rajib Chowdhury

Academic Editor

PLOS ONE